# The Central PXXP Motif Is Crucial for PMAP-23 Translocation across the Lipid Bilayer

**DOI:** 10.3390/ijms22189752

**Published:** 2021-09-09

**Authors:** Sung-Tae Yang, Song-Yub Shin, Sung-Heui Shin

**Affiliations:** 1Department of Microbiology, School of Medicine, Chosun University, Gwangju 61452, Korea; 2Department of Cellular and Molecular Medicine, School of Medicine, Chosun University, Gwangju 61452, Korea; syshin@chosun.ac.kr

**Keywords:** antimicrobial peptide, PMAP-23, PXXP motif, translocation, membrane disruption

## Abstract

PMAP-23, a cathelicidin-derived host defense peptide, does not cause severe membrane permeabilization, but exerts strong and broad-spectrum bactericidal activity. We have previously shown that it forms an amphipathic α-helical structure with a central hinge induced by the PXXP motif, which is implicated in the interaction of PMAP-23 with negatively charged bacterial membranes. Here, we studied the potential roles of the PXXP motif in PMAP-23 translocation across the lipid bilayer by replacing Pro residues with either α-helix former Ala (PMAP-PA) or α-helix breaker Gly (PMAP-PG). Although both PMAP-PA and PMAP-PG led to effective membrane depolarization and permeabilization, they showed less antimicrobial activity than wild-type PMAP-23. Interestingly, we observed that PMAP-23 crossed lipid bilayers much more efficiently than its Pro-substituted derivatives. The fact that the Gly-induced hinge was unable to replace the PXXP motif in PMAP-23 translocation suggests that the PXXP motif has unique structural properties other than the central hinge. Surface plasmon resonance sensorgrams showed that the running buffer almost entirely dissociated PMAP-23 from the membrane surface, while its Pro-substituted derivatives remained significantly bound to the membrane. In addition, kinetic analysis of the sensorgrams revealed that the central PXXP motif allows PMAP-23 to rapidly translocate at the interface between the hydrophilic and hydrophobic phases. Taken together, we propose that the structural and kinetic understanding of the PXXP motif in peptide translocation could greatly aid the development of novel antimicrobial peptides with intracellular targets by promoting peptide entry into bacterial cells.

## 1. Introduction

Antimicrobial peptides (AMPs) provide the first line of host defense and are capable of killing or inhibiting a wide variety of bacterial pathogens [1,2,3]. AMPs have attracted attention as possible alternatives to conventional antibiotics because of their low tendency to develop bacterial resistance [4,5,6,7,8,9,10]. The exact mechanism of AMP’s killing and/or inhibiting actions is not well understood, but it is generally accepted that AMP interacts with the cytoplasmic membrane of the target microbe and affects membrane permeabilization, eventually leading to cell death [11,12,13,14,15]. However, the antibacterial activity of AMPs is not always parallel to their membrane-permeabilizing activity, which suggests that AMPs would have a secondary mechanism of action to eliminate invading pathogens [16,17,18,19]. Indeed, some AMPs exert antimicrobial activity by inhibiting protein and DNA synthesis after entering the bacterial cells [19]. Since the membrane permeabilization and/or cell-penetrating ability of AMPs plays a key role in effective antimicrobial activity, understanding the factors that regulate membrane permeabilization and cell-penetrating properties is of great help in the development of novel antimicrobial drugs [20]. 

Many studies on peptide-lipid interactions using several lipid model systems have been performed to elucidate key aspects of the mode of action of AMP. For example, previous studies have revealed how the hydrophobicity and secondary structure play a crucial role in the process of interaction with lipid bilayers other than the overall net charge for magainin 2, an important class of AMPs [21,22,23]. Recent studies on the interaction between clinically used lipopeptides such as daptomycin and lipid systems have shown that the preferential binding of daptomycin to negatively charged lipids results in lipid phase separation and lipid-packing defects at phase boundaries, eventually leading to bacterial cell death [24,25,26].

Cathelicidins are the largest family of AMPs found in invertebrates. Porcine cathelicidin PMAP-23 exerts rapid and potent microbicidal activity against a broad spectrum of pathogens [18,27,28]. PMAP-23 has a disordered structure in aqueous solution but adopts an amphipathic helix-hinge-helix structure in membranes [29]. We have previously shown that the central hinge region with the PXXP motif plays a structural role in imparting amphipathicity to PMAP-23, which is essential for preferential interaction with the negatively charged membrane and selective antimicrobial activity. At the molecular level, the central PXXP motif allows the initial electrostatic binding of the N-terminal helix to the membrane surface and the insertion of a C-terminal helix into the hydrophobic core of the membrane [18]. Although the PXXP motif is implicated in the interaction of PMAP-23 with the bacterial membrane, the peptide has a relatively low membrane lytic activity, indicating that the bactericidal activity of PMAP-23 may involve mechanisms beyond simply increasing membrane disruption [18]. In addition, it has been reported that PMAP-23 is able to deliver nucleic acids rapidly and efficiently to plasmacytoid dendritic cells [30]. Thus, PMAP-23 may enter the bacterial cell to form a secondary intracellular target. However, no studies have been conducted on the cell-penetrating ability of PMAP-23.

In this study, we investigated whether PMAP-23 translocation across lipid membranes was associated with antimicrobial activity. To examine the roles of the central PXXP motif in the peptide translocation at the molecular and structural level, two Pro residues of PMAP-23 were replaced with either α-helix stabilizing Ala or α-helix destabilizing Gly. We found that Pro to Ala or Gly substitution increased membrane depolarization and permeabilization but decreased antimicrobial activity. Interestingly, we showed that the PXXP motif enables efficient PMAP-23 translocation across the lipid bilayer. Analysis of surface plasmon resonance (SPR) sensorgrams using a two-state reaction model revealed that the PXXP motif could readily dissociate PMAP-23 from the lipid bilayer. The possible role of the central PXXP motif in the antimicrobial mechanism of PAMP-23, including primary permeabilization of bacterial membranes and secondary intracellular targets, is discussed.

## 2. Results and Discussion

### 2.1. Peptide Design and Secondary Structure Analysis

PMAP-23 has an N-terminal helix, a central hinge induced by the PXXP motif, and a C-terminal helix. We have previously shown that the central hinge plays an important role in maintaining the amphipathic structure and exerting strong antimicrobial activity against PMAP-23 [11,13]. To further investigate the structural and functional aspects of the PXXP motif of PMAP-23, two Pro residues of the central hinge were replaced by the α-helix former Ala (PMAP-AA) or α-helix breaker Gly (PMAP-GG) as shown in Figure 1A. The conformation of the peptide analogs was assessed by circular dichroism (CD) spectroscopy in both benign and hydrophobic environments (Figure 1B,C). In the benign buffer, all peptides exhibited a random coiled structure below 200 nm. In the presence of liposomes, PAMP-23 showed negative peaks at 220 nm and 205 nm, which were interpreted in previous studies as being caused by the central flexible region of the peptide [18]. The CD spectrum of PMAP-23 was similar to that observed in PMAP-PG, suggesting that, like Pro, Gly residues also distort the α-helix in the central region. In contrast, PMAP-PA displayed a typical α-helical spectrum, as shown by the negative ellipticity at approximately 208 and 222 nm. As expected, the replacement of Pro with Ala significantly increased α the -helical content (Table 1), suggesting that PMAP-PA probably consists of a single long α-helical structure. Therefore, our results showed that the Pro to Gly substitution retained the central hinge, while the Pro to Ala substitution changed to a continuous α-helical structure by eliminating the central hinge.

### 2.2. Antimicrobial and Hemolytic Activities

PMAP-23 and its analogs were tested for their antibacterial activity against gram-positive *S. aureus* and gram-negative *E. coli* (Table 1). PMAP-23 was quite active against both *S. aureus* and *E. coli* with MIC values in the 2–4 μM range. In contrast, both PMAP-PA and PMAP-PG exhibited approximately four times less potent antimicrobial activity against both *S. aureus* and *E. coli*, with MIC values in the range of 8 to 16 μM. These results suggest that the α-helicity of PMAP-23 is not associated with its potent antibacterial activity. We also evaluated the cytotoxic and hemolytic activity against RAW 264.7 cells and hRBCs, respectively (Table 1). While PMAP-PA showed relatively strong cytotoxicity (37% cytotoxicity at 64 μM) and hemolytic (14% hemolysis at 64 μM) activities, both PMAP-23 and PMAP-PG were nearly inactive. These findings suggest that the central hinge of α-helical peptides is related to cell selectivity between bacteria and mammalian cells. However, the lower antimicrobial activity of PMAP-PG compared to PMAP-23 suggests that a Gly induced central hinge is unable to replace a PXXP motif in exerting potent antimicrobial activity.

### 2.3. Membrane Depolarization by PMAP-23 and Its Analogs

A common feature observed for antimicrobial peptides is their ability to disrupt membrane integrity, leading to the collapse of the transmembrane potential. To determine whether membrane depolarization induced by PMAP-23 and its analogs is related to bacterial death, we assessed their ability to depolarize bacterial cytoplasmic membranes using a potential-sensitive fluorescent probe diSC_3_(5). When diSC_3_(5) was added to a suspension of *S. aureus,* the fluorescence intensity was strongly extinguished and stabilized after approximately 300 s, indicating the accumulation of diSC_3_(5) within the bacterial cytoplasmic membrane (Figure 2). Subsequent addition of peptides increased diSC_3_(5) fluorescence, reflecting membrane depolarization, followed by the addition of gramicidin D to fully dissipate the membrane potential. We found that PMAP-PA with a high degree of α-helicity caused an immediate increase in diSC_3_(5) fluorescence, indicating the rapid collapse of the transmembrane electrochemical gradients of *S. aureus*. PMAP-23 and PMAP-PG exhibited similar abilities to depolarize the bacterial membrane, but were much lower than PMAP-PA. These results suggest that the α-helicity of peptides influences their membrane depolarization activity. However, the lack of correlation between membrane depolarization and antimicrobial activity indicates that loss of membrane potential is not the main bactericidal effect of PMAP-23, and mechanisms may exist.

### 2.4. Membrane Integration of Peptides by Trp Fluorescence Analysis

Lipid composition of the cytoplasmic membrane is very different in eukaryotic and prokaryotic cells. Eukaryotic cell membranes are predominantly composed of zwitterionic phospholipids, such as phosphatidylcholine (PC), whereas bacterial cell membranes are composed largely of negatively charged phospholipids, such as phosphatidylglycerol (PG) [31]. We examined the ability of peptides to bind to electrically neutral PC as well as negatively charged PC/PG (1:1) by monitoring the emission maximum and intensity of Trp fluorescence, which are highly sensitive to the environment (Table 2). In the presence of PC vesicles, only PMAP-PA caused a blue shift (5 nm) in the emission maximum, which explains its relatively strong cytotoxic and hemolytic activity. In contrast, when peptides were added to PC/PG (1:1) liposomes, the Trp emission maxima were shifted to a shorter wavelength (8–9 nm), indicating that all peptides bind to the negatively charged membranes. To further investigate the state of membrane integration of peptides after their interaction with the vesicles, apparent *K*_SV_ values were calculated using a fluorescence quenching experiment with a water-soluble fluorescence quenching agent, acrylamide (Table 2). If the Trp residue is buried in the bilayer, it is less accessible to acrylamide, resulting in a lower *K*_SV_ value. The *K*_SV_ values of peptides were lower in PC/PG (1:1) than in PC vesicles, suggesting that Trp residues of the peptides insert more strongly and deeply into the hydrophobic core of the bacterial membrane. The lower *K*_SV_ value of PMAP-PA over PMAP-23 and PMAP-PG in PC vesicles further explains the direct correlation between its binding to zwitterionic membranes and its cytotoxicity against mammalian cells.

### 2.5. Membrane Disruption by PMAP-23 and Its Analogs

We next investigated the ability of PMAP-23 and its analogs to disrupt membranes by assessing the release of entrapped calcein from vesicles (Figure 3). Consistent with their respective cytotoxic activities against mammalian cells, both PMAP-23 and PMAP-PG elicited no calcein release, whereas PMAP-PA showed strong calcein release with 23% leakage at 50 μM from the PC vesicles. In contrast, all peptides showed similar calcein leakage from the negatively charged PC/PG (1:1) vesicles, which did not correlate with their antimicrobial activity. These results suggest that membrane disruption plays a relatively minor role in the antimicrobial activity of PMAP-23. However, PMAP-23 and PMAP-PG showed much stronger lytic activities toward negatively charged vesicles than zwitterionic ones, indicating that a central hinge may be required for cell selectivity toward bacteria over mammalian cells.

### 2.6. Peptide Translocation across Lipid Bilayers

To determine whether the antimicrobial activity of peptides was related to their intracellular targets, we investigated the translocation of peptides through the lipid bilayer. The ability of the peptide to cross lipid bilayers was assessed by the resonance energy transfer from the Trp residue of the peptides to the dansyl group of phosphatidylethanolamine (PE) incorporated into the vesicles. When peptides translocate across the lipid bilayer, they are broken down by trypsin confined in vesicles, resulting in a loss of fluorescence intensity (Figure 4). We observed a time-dependent decrease in fluorescence intensity over the course of the 600 s recording period, indicating that the peptides readily translocate across the lipid bilayer. In contrast, PMAP-PA and PMAP-PG showed little change in fluorescence, indicating a lack of entry into the vesicles. The ability of peptides to cross the lipid bilayer is consistent with their respective antibacterial activity, suggesting that the cell-penetrating efficiency is a critical factor in determining their antimicrobial potency. Interestingly, a Gly-induced central hinge was unable to replace the PXXP motif in the peptide translocation, suggesting that the PXXP motif has a very specific structural ability other than the central hinge for cell penetration.

### 2.7. Surface Plasmon Resonance (SPR) Analysis

AMPs bind electrostatically to the membrane surface first, then insert themselves into the lipid bilayer via hydrophobic interactions. Finally, SPR was used to investigate how the kinetic interactions of peptides with membranes are related to peptide translocation, since SPR is useful for understanding the kinetic basis for the two-step process. The first step is the binding of peptides to the membrane surface, while the second step corresponds to the insertion of peptides into the hydrophobic core of the lipid bilayer. The SPR sensorgrams consisting of the association and dissociation phases are shown in Figure 5A. A schematic of the two-step interaction of peptides (continuous helix PMAP-PA and helix-hinge-helix PMAP-23) with the lipid bilayers is shown in Figure 5B. Upon injection of the peptide to PC/PG (1:1) membranes immobilized on a Biacore L1 sensor chip, the SPR response unit (RU) was rapidly increased by peptide binding to the membrane surface. When the RU reached almost equilibrium, the buffer flow removed the dissociated peptide. Interestingly, all peptides showed similar RU in the association step, whereas PMAP-23 showed a much lower RU than PMAP-PA and PMAP-PG in the dissociation phase. The higher RU values for PMAP-PA and PMAP-PG suggest that both peptides do not readily separate from the membrane during the dissociation step. By applying the two-state reaction model to the sensorgrams, we estimated the association rate (*k*_a1_, *k*_a2_) and dissociation rate (*k*_d1_, *k*_d2_), as shown in Table 3. The *k*_a1_ and *k*_d1_ values involved in the first step were similar for all peptides. However, PMAP-23 showed significantly different *k*_a2_ and *k*_d2_ values in the second step compared to the other peptides. This result suggests that after binding to the membrane surface, PMAP-23 is more readily inserted into the lipid bilayer and separates from the membrane faster than other peptides. In the second step, the low association and dissociation rate of PMAP-PG suggests that the Gly induced hinge cannot interact with the membrane as fast as the PXXP motif.

### 2.8. Model Elucidating the Translocation Mechanism of PMAP23

A progression model of PMAP-23 translocation across lipid bilayers is illustrated in Figure 5C. Earlier CD and NMR studies showed that PMAP-23 forms a random coil structure in aqueous buffer, while adopting a flexible helix-hinge-helix structure in the lipid bilayer. In the context of the earlier findings on the interaction of PMAP-23 with membranes, the N-terminus of PMAP-23 initially bound to negatively charged lipid headgroups in the outer leaflet, followed by insertion of the C-terminal helix into the membrane. In the present study, we further investigated the potential role of a PXXP motif in PMAP-23 translocation across lipid bilayers. PMAP-23 efficiently migrated from the outer to the inner leaflet of the lipid bilayers. The central hinge induced by a PXXP motif drives conformational flexibility of amphipathic α-helical structures, which may be the driving force for PMAP-23 translocation across the lipid bilayer. However, the fact that the Gly-induced central hinge cannot replace the PXXP motif in efficient translocation suggests that the PXXP motif functions in specific structural capacities such as peptidyl prolyl cis/trans isomerization.

In summary, PMAP23 was bactericidal without causing significant membrane depolarization and permeabilization. Here, we found a correlation between antimicrobial activity and translocation in PMAP-23 and its analogs, and determined that the PXXP motif is a critical structural factor for the cell-penetrating property of PMAP-23. Furthermore, SPR sensorgrams demonstrated that PMAP-23 was rapidly inserted into the hydrophobic lipid core and readily separated from the membrane for translocation through the lipid bilayer. We propose that the central PXXP motif promotes dynamic association and dissociation of PMAP-23 to the lipid bilayer, allowing PMAP-23 to enter bacterial cells. From an experimental point of view, possible strategies that can be adopted to demonstrate the specific mechanism of action and the precise role of the PXXP motif may include optical super-resolution techniques or scanning probe techniques, such as atomic force microscopy. Most importantly, we believe that the PXXP motif can be applied to other AMPs with an amphipathic α-helical structure, which can be of great help in the further development of AMPs with intracellular targets and can generate multifunctional AMPs.

## 3. Materials and Methods

### 3.1. Chemicals, Peptides and Microorganisms

Phospholipids and peptides were purchased from Avanti Polar Lipids Inc. (Alabaster, AL, USA) and Anygen Co. Ltd. (Gwangju, Korea), respectively. Fluorescent probes, including calcein and 3,3-dipropylthiacarbocyanine [DiSC_3_(5)], were purchased from Invitrogen corporation (Carlsbad, CA, USA). Microorganisms were obtained from the Korean Collection for Type Cultures (KCTC). All other reagents were of analytical grade.

### 3.2. Circular Dichroism (CD) Spectroscopy

CD spectroscopy was used to evaluate the secondary structures of the peptides. The CD spectra of peptides were collected using a Jasco J-715 CD spectrophotometer (Jasco Inc., Tokyo, Japan), and their helical contents were determined from ellipticity at 222 nm, as described previously [20,32]. Briefly, the wavelength from 190 to 250 nm was measured at a speed of 50 nm/min, bandwidth of 1 nm, response time, and with a 0.1 nm step resolution. CD spectra were obtained in the presence and absence of PC/PC (1:1) liposomes (2 mM) at a peptide concentration of 25 μM.

### 3.3. Antimicrobial Activity

The antimicrobial activity of the peptides was tested against gram-positive *Staphylococcus aureus* and gram-negative *Escherichia coli* by a broth microdilution method, as described previously [20]. Briefly, a colony of bacteria was incubated in culture medium (10 mL) and incubated at 37 °C for 14–18 h. After an aliquot (50 μL) of this culture was transferred to fresh culture medium (10 mL), each bacterial strain was grown to reach the mid-logarithmic phase. A total of 100 μL of bacteria (2 × 10^6^ CFU/mL) were added to 96-well microtiter plates in the presence of serial dilutions of peptides. The minimal inhibitory concentration (MIC) was expressed as the lowest concentration of peptide that completely inhibited bacterial growth. Measurements were repeated at least three times in triplicate.

### 3.4. Cytotoxic and Hemolytic Activity

Cytotoxic activity against RAW264.7 cells was measured using the 3-(4,5-dimethylthiazol-2-yl)-2,5-diphenyltetrazolium bromide (MTT) assay, as described previously [33]. Briefly, RAW264.7 cells were incubated under 5% CO_2_ at 37 °C for 24 h in Dulbecco’s modified Eagle’s medium containing 10% fetal calf serum. They were then seeded in 96-well plates (2 × 10^4^ cells/well). Next, peptides were added to the plate to react with the cells for 48 h. MTT (20 μL) was added to the plate and incubated for 4 h at 37 °C; the resulting MTT formazan precipitate was then dissolved in DMSO. The absorbance at 550 nm was measured using an enzyme-linked immunosorbent assay (ELISA) plate reader (Molecular Devices, Sunnyvale, CA, USA). The hemolytic activity against human red blood cells (hRBCs) was evaluated by monitoring the release of hemoglobin by incubation with peptides, as described previously [34]. Approximately 10^6^ hRBC/mL were washed twice with 35 mM phosphate buffer (150 mM NaCl, pH 7.2). Then, 100 μL of 4% (*v*/*v*) hRBCs suspended in the buffer was added to a 96-well plate in the presence of peptides, followed by incubation at 37 °C for 1 h. The plates were centrifuged at 1000× *g* for 5 min, and the supernatant was transferred to clean 96-well plates. We measured the absorbance at 414 nm for hemoglobin release using an ELISA microplate reader. Zero and complete (100%) hemolysis was achieved in PBS and 0.1% Triton X-100, respectively. The percent cytotoxic and hemolytic activity of the peptides was the mean of triplicate measurements in three independent assays.

### 3.5. Membrane Depolarization

The assay for depolarization of the cytoplasmic membrane was performed using the potential-sensitive fluorescent probe diSC3(5), as described previously [35,36]. Briefly, the *S. aureus* strains grown to the logarithmic phase at 37 °C were resuspended in HEPES buffer (100 mM KCl, 20 mM glucose, pH 7.2) to an optical density at 600 nm (OD600) of 0.05. Fluorescence was monitored using an RF-5301 spectrofluorometer (Shimadzu, Tokyo, Japan) at an emission wavelength of 670 nm (excitation wavelength = 622 nm). Complete dissipation of *S. aureus* membrane potential was determined by the addition of gramicidin D.

### 3.6. Preparation of Vesicles

Small unilamellar vesicles (SUVs) were generated by sonication for CD, Trp fluorescence, and SPR experiments. Phospholipids consisting of PC or PC/PG (1:1) dissolved in chloroform were dried in the presence of nitrogen gas. The lipid film on the walls of a glass tube was further dried under vacuum overnight, and then resuspended in sodium phosphate buffer (10 mM) while vortexing. A titanium-tipped sonicator was applied to the suspension in an ice bath until the suspension became clear. Large unilamellar vesicles (LUVs) were generated by extrusion for calcein leakage and translocation experiments. The lipid film was hydrated with an appropriate buffer with or without 70 mM calcein. The suspension was subjected to five freeze-thaw cycles and then successively extruded through a polycarbonate filter with a pore size of 100 nm. Free calcein was removed from calcein-entrapped LUVs using a Sephadex G-50 column.

### 3.7. Calcein Leakage from LUVs

Membrane permeabilization by peptides was assessed by calcein release from LUVs. The fluorescence intensities were measured 10 min after the addition of peptides at 520 nm with an excitation wavelength of 490 nm on a Shimadzu RF-5301 spectrofluorometer. The percentage of calcein leakage caused by the peptides was calculated using the following equation: calcein leakage (%) = 100 × (F − F_0_)/(F_t_ − F_0_), where F_0_ is the initial fluorescence observed without the peptides, F is the fluorescence intensity in the presence of peptides, and F_t_ is the fluorescence achieved by adding Triton X-100. The measurements were repeated three times under each condition.

### 3.8. Tryptophan (Trp) Fluorescence

Peptide binding to membranes was estimated using Trp fluorescence measurements, as described previously [34]. Briefly, each peptide was added to vesicles at a peptide/lipid molar ratio of 1:100, and the mixture was allowed to interact for 10 min. The Trp emission spectra of peptides were recorded at 300–400 nm, and Trp fluorescence quenching was performed by adding acrylamide. The Stern-Volmer (*K*_SV_) constant was calculated using the following equation: *K*_SV_[Q] = F_0_/F − 1, where [Q] is the concentration of acrylamide, F_0_ is the Trp fluorescence in the absence of acrylamide, and F is the Trp fluorescence in the presence of acrylamide. The measurements were repeated in triplicate at a given condition to ensure reproducibility.

### 3.9. Peptide Translocation

Peptide translocation across a phospholipid bilayer was evaluated by fluorescence resonance energy transfer (FRET) from the Trp of peptides to a 1-dimethylaminonaphthalene-5-sulfonyl chloride (dansyl or DNS) group of phosphatidylethanolamine (PE), as described previously [37,38]. Briefly, after lipid film composed of PC/PG/DNS-PE (50:45:5) was hydrated with 20 mM HEPES buffer (150 mM NaCl, pH 7.2) containing 200 μM trypsin, liposomes with encapsulated enzymes were prepared by extrusion. To inactivate the trypsin outside the liposomes, 2000 μM trypsin inhibitor was applied to the vesicle suspension. For the control experiment, vesicles encapsulated with an enzyme inhibitor were also used. The excitation at 280 nm of Trp residues caused FRET in the DNS-PE, and the emission at 510 nm of the dansyl group was recorded. Fluorescence changes caused by the addition of peptides (1 μM) were observed using a Shimadzu RF 5301 PC spectrofluorometer. The fluorescence reduction by digestion of the internalized peptide indicates peptide translocation across the lipid bilayers. Measurements were performed in triplicates.

### 3.10. Kinetic Analysis of Molecular Interaction Using Surface Plasmon Resonance (SPR)

The kinetics of peptide binding to and dissociation from the lipid bilayer were assessed by SPR using BIACORE 2000, as described previously [34]. We injected vesicles into the L1 sensor chip at a flow rate of 2 μL/min for 15 min and applied sodium hydroxide (10 mM) at a flow rate of 50 μL/min to remove any possible lipid multilamellar structures from the surface. The obtained sensorgrams of the peptides were globally fitted using BIA evaluation software. We analyzed the sensorgrams using a two-state reaction model to evaluate the association and dissociation rate constants. This model describes membrane binding and insertion of peptides, which, with regard to peptide-lipid interaction, may correspond to P + L $\begin{array}{c} k_{a1}\\ ↔\\ k_{d1} \end{array}$ PL $\begin{array}{c} k_{a2}\\ ↔\\ k_{d2} \end{array}$ PL* where in the first step, P (peptide) initially binds to L (lipid) to produce the PL complex as a result of initial binding. The change in PL to PL* in the second step corresponds to peptide insertion into the hydrophobic core of the lipid bilayer.

## Figures and Tables

**Figure 1 ijms-22-09752-f001:**
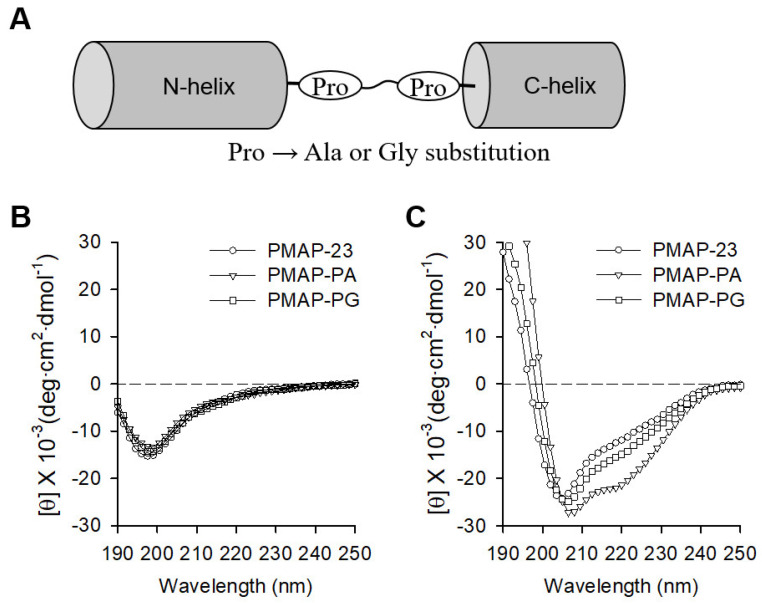
Secondary structures of PMAP-23 and its analogs. (**A**) Schematic diagram of PMAP-23 with a helix-hinge-helix structure. Two Pro amino acids of PMAP-23 were substituted with Ala (PMAP-PA) or Gly (PMAP-PG). (**B**,**C**) CD spectra of PMAP-23, PMAP-PA, and PMAP-PG. Spectra were recorded at a peptide concentration of 20 μM in aqueous buffer (**B**) and in 2 mM PC/PG (1:1) liposomes (**C**).

**Figure 2 ijms-22-09752-f002:**
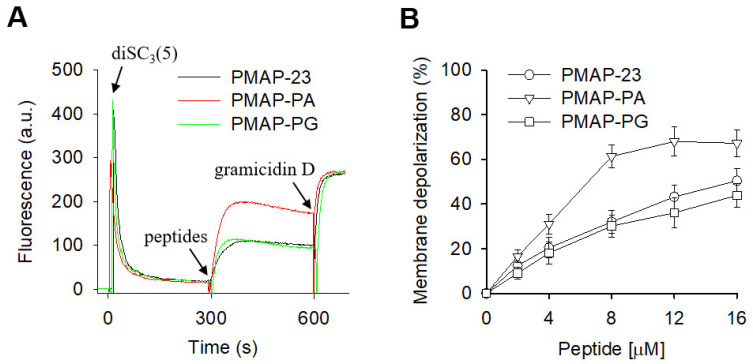
Depolarization of bacterial membranes by peptides. (**A**) Time course of florescence intensity of the membrane potential-sensitive probe diSC_3_(5). When the diSC_3_(5) was added to suspensions of *Staphylococcus aureus*, the fluorescence was stabilized after about 300 s, and peptides (8 μM) were subsequently added. (**B**) Dose-dependent peptide-induced membrane depolarization of *S. aureus*. Complete membrane depolarization was determined by addition of Gramicidin D and zero levels correspond to fluorescence before peptide addition.

**Figure 3 ijms-22-09752-f003:**
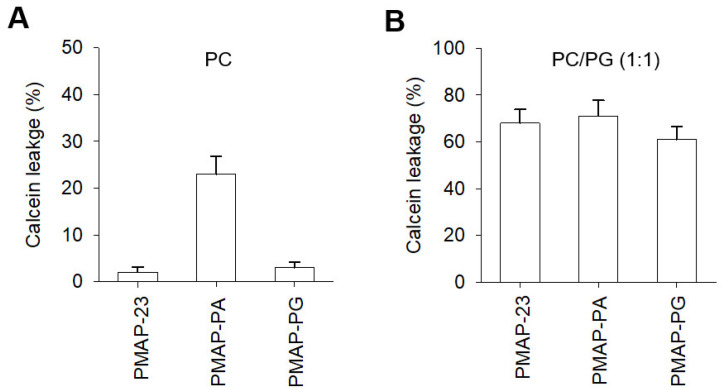
Ability of peptides to induce calcein leakage from (**A**) PC and (**B**) PC/PG (1:1). Peptides were added at 50 μM for 100 μM PC and 10 μM for 100 μM PC/PG.

**Figure 4 ijms-22-09752-f004:**
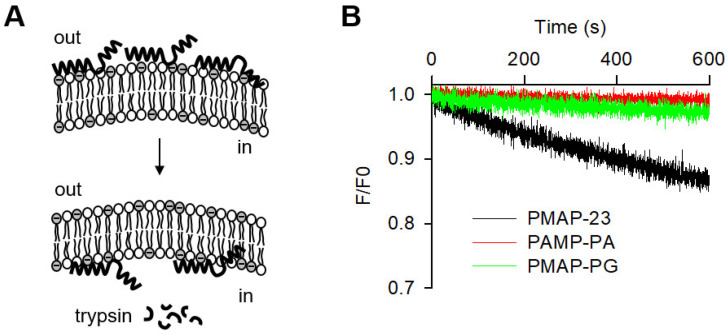
Translocation of the peptides across lipid bilayers. (**A**) Schematic diagram of the experiment. Peptides were exposed to vesicles containing trypsin and the internalized peptide was digested by liposome-entrapped trypsin. (**B**) Time-dependent peptide translocation. A reduction in fluorescence intensity after the addition of a peptide is indicative of digestion of the internalized peptide.

**Figure 5 ijms-22-09752-f005:**
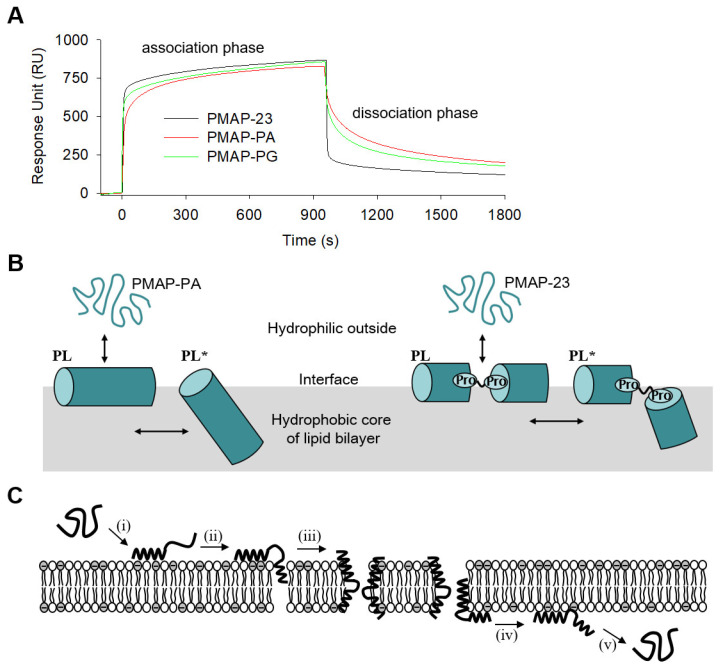
Sensorgrams for interaction of PMAP-23 and its analogs with membranes. (**A**) Upon injection of peptides at time = 0 s, they were associated with PC/PG (1:1) lipid bilayers immobilized on the L1 sensor chip surface; phosphate buffer was added to initiate dissociation at time = 960 s. (**B**) Schematic comparison of the interaction of continue helix (PMAP-PA) and helix-hinge-helix (PMAP-23) with lipid bilayers. (**C**) Schematic overview of PMAP-23 translocation across lipid bilayers: (i) association of the N-terminal α-helix to the outer leaflet; (ii) insertion of the C-terminal α-helix into the core of lipid bilayers; (iii) peptide-induced pore formation in lipid bilayers; (iv) peptide translocation to the inner leaflet; (v) dissociation of the peptide from membranes.

**Table 1 ijms-22-09752-t001:** Primary structure, helicity, and antimicrobial and cytotoxic activity of PMAP-23 and its analogs.

Peptides	Amino Acid Sequences	Helicity ^a^	MIC (μM)		Cytotoxicity ^b^	Hemolysis ^c^
			*S. aureus*	*E. coli*		
PMAP-23	RIIDLLWRVRRPQKPKFVTVWVR	27%	2–4	2–4	0%	0%
PMAP-PA	RIIDLLWRVRRAQKAKFVTVWVR	63%	8–16	8–16	37%	14%
PMAP-PG	RIIDLLWRVRRGQKGKFVTVWVR	31%	8	8–16	3%	0%

^a^ Contents of α-helices were estimated from the CD spectra shown in Figure 1. ^b^ Cytotoxic activity against murine RAW 264.7 cells was performed at 64 μM. ^c^ Hemolytic activity against human red blood cells was performed at 64 μM.

**Table 2 ijms-22-09752-t002:** Trp fluorescence emission maxima (λ_max_) and Stern-Volmer constants (*K*_SV_) for PMAP-23 and its analogs in the presence of vesicles composed of PC or PC/PG (1:1) at a peptide/lipid molar ratio of 1:100.

Peptide	PC		PC/PG (1:1)	
	λ_max_	*K* _SV_	λ_max_	*K* _SV_
PMAP-23	352 nm (0 nm) ^a^	15.26	343 nm (9 nm)	7.55
PMAP-PA	347 nm (5 nm)	11.73	343 nm (9 nm)	7.17
PMAP-PG	351 nm (1 nm)	15.12	344 nm (8 nm)	7.42

^a^ The numbers in parentheses indicate the blue shift of the emission maxima compared to the aqueous buffer.

**Table 3 ijms-22-09752-t003:** Association (*k*_a1,_
*k*_a2_) and dissociation (*k*_d1,_
*k*_d2_) kinetic rate constants for PMAP-23 and its analogs interacting with PC/PG (1:1) by a two-state reaction model.

Peptide	First Step		Second Step	
	*k*_a1_ (1/Ms)	*k*_d1_ (1/s)	*k*_a2_ (1/s)	*k*_d2_ (1/s)
PMAP-23	164	8.7 × 10^−3^	82.3 × 10^−3^	732.5 × 10^−3^
PMAP-PA	375	9.0 × 10^−3^	2.1 × 10^−3^	73.8 × 10^−3^
PMAP-PG	284	9.4 × 10^−3^	2.7 × 10^−3^	68.2 × 10^−3^

## Data Availability

Not applicable.

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
