# Peer review of "The Central PXXP Motif Is Crucial for PMAP-23 Translocation across the Lipid Bilayer"

_ijms, 2021, doi:10.3390/ijms22189752_

Round 1

Reviewer 1 Report

The article by S.Yang et al. proposes an experimental study about the interaction between PMAP-23, a cathelicidin-derived host defense peptide, and lipid bilayers with different structure and composition. In particular, the role of PXXP motif has been investigated by replacing Pro a.a. residue with Ala and Gly residues, respectively. PA and PG modified forms result structurally different (single alpha-helix for PA and helix-hinge-helix for PG) allowing to get information about several crucial aspects as, i.e., the role of secondary structure with regard to antimicrobial and hemolytic activity, the membrane depolarization/disruption and the peptide translocation across lipid bilayer, all aspects useful in the understanding of the mechanism of action which regulates the interaction. The pivotal role of PXXP motif has been demonstrated by different experimental techniques including circular dichroism, surface plasmon resonance and several fluorescent spectroscopies.

The manuscript is clear, well-written and well-organized. The experimental procedures are systematic and supported by the statistics. Figures are clear and appealing, discussions are well-founded and supported. The results are interesting with a high potential of disruptive applications. However, the discussion, especially in the introductive part, can be broaden to emphasize some key aspects. Recommendations to improve the quality of the manuscript are listed below:

- In light of the several lipid model system used, the beneficial contribution of the studies about the peptide-lipid interaction to shed light on mechanisms of action should be highlighted. For example, some quite recent studies revealed how even in the case of magainin2, another important class of AMPs, the hydrophobicity and secondary structure play a crucial role in the process of interaction with lipid bilayers other than the overall net charge. Similarly, the interaction between clinically used lipo-peptide as Daptomicyn (Cubicin) and lipid systems have been studied evaluating the role of lipid composition, the presence of phase separation and eventually the asymmetry of lipid leaflet. The addition of a short topic overview in the introduction can help not only in broadening the audience but also to raise awareness of the importance of peptide engineering in the detection of AMPs mechanism of action and how it can be inspired by the understanding of the relationship between AMPs property and chemical modification. Authors can critically use the following articles: (BBA-Biomembranes 1860 (2018) 2635-2643 (https://doi.org/10.1016/j.bbamem.2018.10.003); Colloids and Surface B: Biointerfaces 2018, 432-440 (https://doi.org/10.1016/j.colsurfb.2018.04.034); IJCS 2019, 247-258 (https://doi.org/10.1016/j.jcis.2019.06.028), J. Phys. Chem. B 2020, 124, 39, 8562–8571 (https://doi.org/10.1021/acs.jpcb.0c06640), BBA-Biomembranes 2020, 183395 (https://doi.org/10.1016/j.bbamem.2020.183395) BBA-Biomembranes 2006, 1292-1302 (https://doi.org/10.1016/j.bbamem.2006.02.001), J. Phys. Chem. B 2021, 125, 22, 5775–5785  https://doi.org/10.1021/acs.jpcb.1c02047).

-at line 152 authors state " only PMAP-23 caused a blue shift (5 nm)" instead of PMAP-PA; please correct.

-Figure 3 would requires labels as PC and PC:PG (1:1) over the panels to help the reading.

-At line 221 authors state " The sensorgrams of PMAP-PA and PMAP-PG did not return to zero, ". Actually, neither the one of PMAP-23 returns to zero. Please, rephrase.

-At line 224 authors state " The ka1 and kd1 values involved in the first step were similar for all peptides. " Actually, ka1 of PMAP-23 is lower than PA form and also the value of PMAP-PG is slightly lower than PA form sign that even the first step could be promoted by the presence of the hinge. Please, add comments.

-At lines 253-255 authors state " However, the fact that the Gly induced central hinge cannot replace the PXXP motif in efficient translocation suggests that the PXXP motif functions in specific structural capacities such as peptidyl prolyl cis/trans isomerization."  Can you add reference?

- A dedicated conclusion section is missing; it could be simply obtained separating the text from line 257 but considerations on possible strategies to be adopted to reveal the specific mechanism of action and the precise role of PXXP motif should be added. For example, from an experimental point of view, optical super-resolution techniques or scanning probe techniques, such as AFM, could help in revealing crucial aspects of the mechanism of action. Some comments about it would improve the quality of the manuscript.

Reviewer 2 Report

The manuscript by Yang et. al. describing structural studies of PMAP-23 is an interesting piece of work and I do not find major flaws in the experiments and/or results presented. However, this reviewer feels that the paper would add much more value if the membrane penetration and translocation experiments may be reproduced using living cells in culture. Although liposomes are accepted and widely used as model membranes to study the biophysical aspects of membrane interactions, a study that claims applications of AMP in its antibacterial activities should present data using living cells. 

Round 2

Reviewer 1 Report

Authors addressed almost all the points raised up by the reviewer